# Research on sports activity behavior prediction based on electromyography signal collection and intelligent sensing channel

Fengjin Ye[1], Yuchao Zhao[2] and Zohaib Latif[3]

[1] College of Ocean Culture and Tourism, Xiamen Ocean Vocational College, Xiamen, Fujian, China
[2] College of Information Engineering, Xiamen Ocean Vocational College, Xiamen, China
[3] Department of Computer Science, School of Engineering and Digital Sciences (SEDS), Nazarbayev University, Astana, Kazakhstan

## ABSTRACT

Sports behavior prediction requires precise and reliable analysis of muscle activity during exercise. This study proposes a multi-channel correlation feature extraction method for electromyographic (EMG) signals to overcome challenges in sports behavior prediction. A wavelet threshold denoising algorithm is enhanced with nonlinear function transitions and control coefficients to improve signal quality, achieving effective noise reduction and a higher signal-to-noise ratio. Furthermore, multi-channel linear and nonlinear correlation features are combined, leveraging mutual information estimation *via* copula entropy for feature construction. A stacking ensemble learning model, incorporating extreme gradient boosting (XGBoost), K-nearest network (KNN), Random Forest (RF), and naive Bayes (NB) as base learners, further enhances classification accuracy. Experimental results demonstrate that the proposed approach achieves over 95% prediction accuracy, significantly outperforming traditional methods. The robustness of multi-channel correlation features is validated across diverse datasets, proving their effectiveness in mitigating channel crosstalk and noise interference. This work provides a scientific basis for improving sports training strategies and reducing injury risks.

## INTRODUCTION

In sports research, scholars have always attached great importance to studying the mechanism of sports fatigue, but so far, there is no recognized theory for the mechanism of sports fatigue. Over the years, numerous sports researchers have proposed hypotheses with a theoretical basis; however, these hypotheses fail to explain the mechanism of sports fatigue adequately. The application of electromyography (EMG) signals in sports is pervasive, which helps to gain a deeper understanding of muscle movement patterns and fatigue levels, improve sports skills, and monitor sports injuries (*Al-Ayyad et al., 2023*). In competitive sports, athletes usually improve their performance by increasing training



Corresponding author
Fengjin Ye, yefengjin@xmoc.edu.cn

intensity, but if the exercise intensity is not scientifically controlled, it can easily lead to fatigue and muscle injury. After high-intensity training, the human body may experience a decline in muscle function, known as muscle fatigue. The occurrence of muscle fatigue not only affects athletic performance but can also lead to long-term muscle damage in severe cases. Therefore, using scientific and technological means to predict athletes' muscle fatigue status and adjust training intensity reasonably has important practical application value (*Tan & Ran, 2023*).

The changes in EMG signals can effectively reflect muscle fatigue status. However, as weak bioelectric signals, EMG signals are easily affected by external factors such as power frequency interference and electronic device noise during the acquisition process, resulting in valuable information being overwhelmed by noise (*Cheng et al., 2023a*). Therefore, effective denoising methods must be adopted to improve the reliability of the signal. Traditional denoising methods such as Kalman filtering and spectral subtraction have limited effectiveness in processing noisy EMG signals and cannot fully remove noise. In response to this issue, this article proposes a denoising method based on wavelet thresholding, which has been widely applied in weak electrical signal processing and image denoising.

On the other hand, due to the coupling phenomenon between muscles during movement, electrodes placed on the target muscle may record activity signals from nearby muscles, resulting in crosstalk between different channels (*Wang et al., 2023*). This article proposes an EMG signal recognition method based on multi-channel correlation (MC) features to explore the information on EMG signals between multiple channels. Specifically, the research work of this article includes the following aspects: Firstly, the application of wavelet threshold denoising algorithm in arm EMG signal detection was studied. In response to the problems of discrete points and large errors in traditional algorithms, nonlinear function transition and control coefficients were introduced to improve the threshold denoising algorithm. The improved algorithm can effectively improve the accuracy and reliability of signal recognition while removing noise. This article proposes an EMG signal recognition method based on multi-channel correlation features. Extract linear correlation features between channels by calculating the consistency correlation coefficient between channels; Meanwhile, utilize mutual information to obtain nonlinear correlation features between channels. To improve the accuracy and efficiency of mutual information estimation, this article introduces a mutual information estimation method based on copula entropy, which avoids the joint probability density estimation problem.

## Major contributions

The main contributions of this article are as follows:

In response to the shortcomings of soft and hard threshold denoising methods, this article introduces nonlinear functions and control coefficients to improve the wavelet threshold denoising algorithm, significantly enhancing the denoising effect of electromyographic signals and improving the accuracy and stability of subsequent signal recognition.

A method for identifying EMG signals based on multi-channel correlation features is proposed, which combines the linear and nonlinear correlation characteristics between channels. The mutual information estimation based on copula entropy effectively solves the channel crosstalk problem and improves signal recognition's reliability.

# RELATED WORKS

This article mainly introduces related work from three aspects: EMG signal denoising, feature selection, and classifier integration.

## EMG signal denoising

As an extension of the Fourier transform, wavelet analysis has been widely studied in signal denoising (*Song et al., 2023*). Wavelet transform (WT) is a multi-resolution analysis tool suitable for non-stationary and fast transient signals, particularly for studying neurophysiological signals. *Pan et al. (2024)* proposed a threshold method based on wavelet denoising, which decomposes electromyographic signals into wavelets and analyzes wavelet coefficients using a weighted average of soft and hard thresholds. Finally, the denoised electromyographic signals are reconstructed, retaining useful information and effectively removing noise. *Qi et al. (2025)* proposed a new signal-to-noise ratio (SNR) estimator that evaluates signal reconstruction quality through wavelet decomposition and denoising, verifying the effectiveness of wavelet denoising on EMG signals. *Houssein et al. (2024)* compared four classic thresholding algorithms: general thresholding, heuristic thresholding, mixed thresholding, and minimum-maximum thresholding. The results showed that the general and soft thresholding functions performed the best, and the second-order Daubechies wavelet was slightly better than other algorithms. *Yousuf Mir & Singh (2024)* proposed a new method for denoising electrocardiogram signals damaged by Gaussian white noise using wavelet packet transform and soft thresholding, which outperforms traditional methods in terms of performance. *Li et al. (2024)* proposed a time-frequency analysis method for EMG signals based on wavelet transform, which outperforms traditional Fourier methods in handling discontinuous and spike signals.

### EMG feature selection

Traditional methods for extracting EMG signals often extract many different features, resulting in high feature dimensions and redundancy, affecting the model's accuracy of the data analysis. Feature selection methods can effectively filter out useful features and remove irrelevant and redundant parts. Depending on the way features are evaluated, feature selection methods can be divided into three categories: Filter (*Sîmpetru et al., 2024*), Wrapper (*Shi et al., 2023*), and Embedded methods (*Ariyanto et al., 2023*).

The Filter method evaluates the quality of features based on the inherent properties of the data, usually by statistically measuring and analyzing the correlation between features and categories (*Solorio-Fernández, Carrasco-Ochoa & Martínez-Trinidad, 2024*). A univariate filter employs a single statistical metric, such as information gain or ReliefF (*Begum et al., 2024*), to evaluate and rank features. In contrast, a multivariate filter considers feature dependencies by analyzing the entire feature subset for selection (*Esfandiari, Khaloozadeh & Farivar, 2023*). The wrapper method combines machine

learning models and uses model performance evaluation as the criterion for feature selection (*Alija et al., 2023*). *Nouri-Moghaddam, Ghazanfari & Fathian (2023)* indicates that the Wrapper method is usually superior to the Filter method because the Filter ignores the specific performance of the selected features in the classification algorithm. The Embedded method combines the advantages of Filter and Wrapper. It simultaneously searches for the optimal feature set when constructing the classifier based on optimizing the objective function for feature selection (*Zhao et al., 2023*).

## Classifier construction

After extracting appropriate features, it is necessary to train a classifier to establish a correlation model between feature vectors and gesture actions to recognize EMG signal features (*Prabhavathy, Elumalai & Balaji, 2024*).

With the rapid development of deep learning, more and more researchers are applying neural networks to electromyographic signal recognition. Convolutional neural networks (CNNs) and recurrent neural networks (RNNs) are commonly used neural network methods in gesture recognition (*Robinson et al., 2023*). *Wei et al. (2019)* compared the performance of RNN and feedforward neural network (FFNN) in gesture recognition, and the experiment showed that RNN with fewer parameters can achieve similar accuracy as FFNN in a shorter running time (*Simão, Neto & Gibaru, 2019*). In addition, some researchers combine CNN with RNN to capture temporal and spatial features fully. *Hu et al. (2018)* proposed a hybrid CNN-RNN network based on an attention mechanism in which the upper layer is a multi-layer CNN, and the lower layer is a long short-term memory (LSTM) network, which can effectively extract spatiotemporal features.

# METHODOLOGY

## Surface electromyography signal

The processing method of surface EMG signals is an important part of EMG signal analysis and application, and the choice of processing method primarily depends on the mechanism of EMG signal generation. There are four main mathematical models for surface electromyography signals: linear system model, bipolar model, lumped parameter model, and non-stationary model. The first three models all belong to steady-state analysis when the muscle exerts constant force in a steady state. When muscles exert uneven force, the changes in EMG signals are unstable, and non-stationary models can be used to characterize EMG signals.

$$y(t) = c(t)m(t) \tag{1}$$

where c(t) is the modulation signal of muscle contraction degree and m(t) is the Gaussian noise (zero mean unit variance) carrier signal.

The original surface electromyography signal is non-stationary and susceptible to environmental noise and other bioelectric signals (such as electrocardiogram). It is

necessary to denoise the original surface electromyography signal. The proper frequencies of EMG signals are concentrated between 5 and 200 Hz. In this study, the wavelet threshold denoising method was used to filter out the noise components in the signal.

The commonly used wavelet filtering methods include Bayesian and non-Bayesian methods. Non-Bayesian methods can be divided into three types: (1) spatial correlation filtering algorithm, (2) maximum modulus reconstruction filtering algorithm, and (3) wavelet threshold filtering. By analyzing and comparing three filtering methods, it is concluded that the wavelet threshold filtering algorithm has the characteristics of simple real-time operation and low computational complexity. Therefore, this article conducts research on the wavelet threshold denoising method.

Let the basic wavelet be $\Psi_{m,n}(t)$ and $f(t)$ be a square integrable function, as shown in Eq. (1):

$$\Psi_{m,n}(t) = \frac{1}{\sqrt{m}} \Psi\left(\frac{t-n}{m}\right). \tag{2}$$

The continuous wavelet transform of signal $f(t)$ is defined as:

$$W_{\Psi_f}(m,n) = \frac{1}{\sqrt{m}} \int_{-\infty}^{\infty} f(t) \Psi^*\left(\frac{t-n}{m}\right) \mathrm{d}t \tag{3}$$

where $m > 0$, $n \in \mathbf{R}$, m and n are scale factor and displacement, respectively.

If the basic wavelet $\Psi_{m,n}(t)$ satisfies the following permissible conditions:

$$C_j = \int_R |\omega|^{-1} \left|\hat{\Psi}_{(\omega)}\right|^2 \mathrm{d}\omega \tag{4}$$

where $C_j$ is the wavelet transform coefficient.

The continuous wavelet transform is

$$f(t) = \frac{1}{C_j} \int_{-\infty}^{\infty} \int_{-\infty}^{\infty} \frac{1}{m^2} W_{\Psi_f}(m,n) \Psi_{m,n}(t) \mathrm{d}m \mathrm{d}n. \tag{5}$$

Through research and analysis, it has been found that after wavelet transform, $C_j$ generates a lot of redundancy. Therefore, denoising and restoring noisy signals can utilize this redundancy. When certain conditions are met, the basis function that can form a square integrable function with $f(t)$ can be represented by a set of baselines in the square-integrable function. After theoretical analysis, it is found that the wavelet transform generates bases in a simple manner.

## Based on improved threshold algorithm
### Threshold-based wavelet denoising method

The main research content of this article is to improve the denoising effect of signals by optimizing the selection threshold during the denoising process. The threshold-based

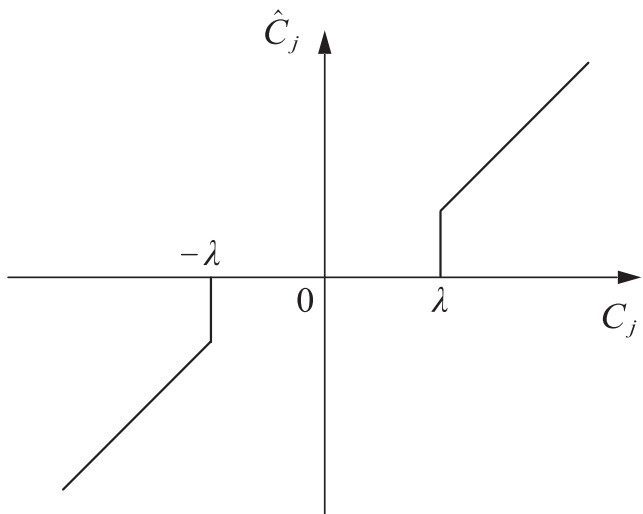

**Figure 1 Hard threshold function.**

selection method includes two types, soft threshold and hard threshold, which are introduced as follows:

(1) Hard threshold function analysis. The hard threshold function is shown in Eq. (6)

$$\hat{C}_j = \begin{cases} C_j, & |C_j| \geq \lambda \\ 0, & |C_j| < \lambda \end{cases} \tag{6}$$

where $\lambda$ is the set threshold, and $\hat{C}_j$ is the wavelet coefficient after threshold filtering. When the wavelet coefficient $C_j$ is less than $-\lambda$ or greater than $\lambda$, this part of the wavelet coefficient is recognized as the wavelet coefficient of the original signal. On the contrary, the wavelet coefficients are 0.

The hard threshold wavelet denoising method can obtain reconstructed signals with strong similarity. However, in this method, the signal edge is prone to generating oscillation thresholds, forming abrupt signals. The function curve is shown in Fig. 1.

(2) Soft threshold function. The soft threshold function is shown in Eq. (7)

$$\hat{C}_j = \begin{cases} C_j - \lambda, & C_j \geq \lambda \\ 0, & |C_j| < \lambda \\ C_j + \lambda, & C_j \leq -\lambda. \end{cases} \tag{7}$$

In Eq. (7), when $C_j$ is greater than $\lambda$, subtract $\lambda$ from the wavelet coefficients; When $C_j$ is less than $-\lambda$, the wavelet coefficients plus $\lambda$ can be used as reconstruction objects; When $C_j$ is between $-\lambda$ and $\lambda$, filter out this part of the wavelet coefficients, and the processed wavelet coefficients are 0. By using wavelet soft thresholding denoising method, reconstructed signals with good smoothness can be obtained. However, the threshold set in this method is biased, which can cause the filtered signal to sometimes be too smooth, resulting in significant errors. The function curve is shown in Fig. 2.

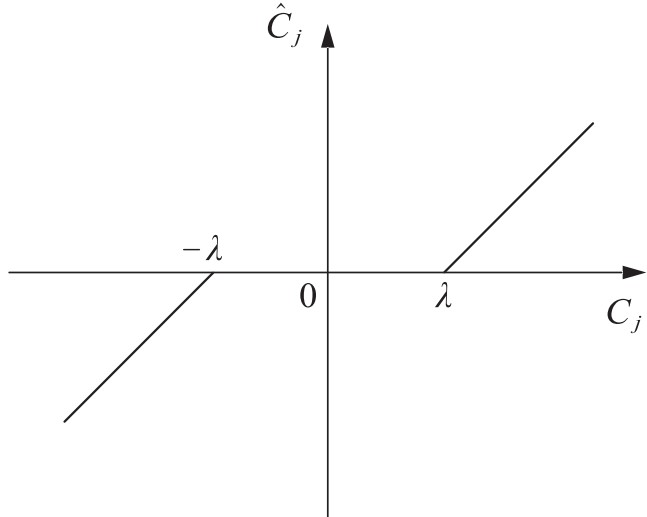

**Figure 2 Soft threshold function.**

## Analysis of improving threshold denoising method

In the initial stage of denoising EMG signals, although the hard thresholding algorithm can obtain reconstructed signals with strong similarity, due to the fluctuation of its threshold function, it will produce many abrupt noise points in signals containing multiple oscillations. At this time, the filtered signal will exhibit many distortion phenomena. Although the soft thresholding algorithm function is continuous, it appears relatively smooth when processing the wavelet coefficients of oscillating signals, which largely overcomes the shortcomings of the hard thresholding algorithm. Similarly, this algorithm significantly impacts wavelet coefficients with large absolute values. It can also cause loss of high-frequency component information in the signal, resulting in overly smooth processed signals and blurred signal edges, which can also increase errors in the reconstructed signal. Therefore, by introducing a nonlinear function and control coefficients during the transition phase of the function, threshold parameters can be set using such a threshold function.

Based on the above analysis, this article adopts a nonlinear function for transition and introduces control coefficients. The expression of the improved threshold algorithm is shown in Eq. (8).

$$
\hat{C}_j =
\begin{cases}
C_j - \lambda + ae^\lambda - a, & C_j \geq \lambda \\
a|e^{C_i} - 1|, & -\lambda \leq C_j < \lambda \\
C_j + \lambda - ae^{-\lambda} + a, & C_j \leq -\lambda
\end{cases}
\tag{8}
$$

where a is the introduced control coefficient, which affects the degree of variation of wavelet coefficient $C_j$ in different ranges, the function curve is shown in Fig. 3.

The improved threshold function proposed in this article is based on the soft threshold function and introduces control coefficients to make the original signal and noise signal excessively smooth, and the filtered signal is closer to the original signal. This method has a simple idea and is conducive to improving the denoising effect.

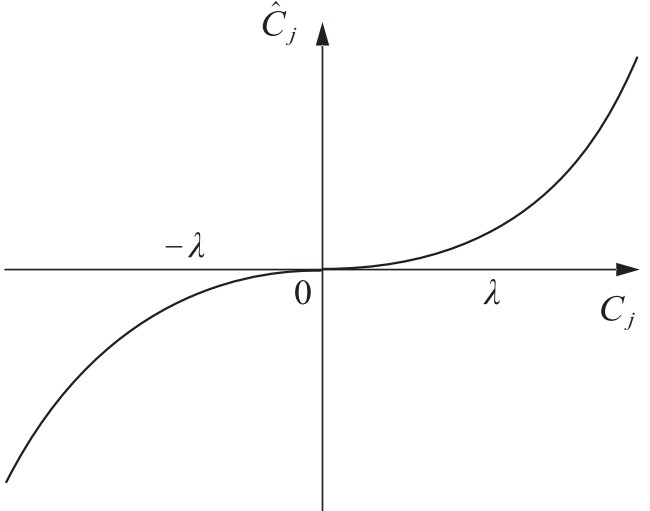

**Figure 3** **Improve the threshold function.**

## Multi-channel correlation characteristics of EMG signals

When analyzing the correlation information between channels of EMG signals, traditional methods of measuring correlation usually only describe the linear correlation between two channels and rarely consider nonlinear correlation, making it difficult to represent the characteristics of nonlinear EMG signals accurately. This article calculates the linear correlation features between channels, estimates the nonlinear correlation features, and combines the two to construct multi-channel correlation features for pose recognition.

### Linear correlation features between channels

In *Fraschini et al. (2019)*, a calculation method for inter-channel consistency correlation coefficient (CCC) is defined to measure the consistency relationship between two variables. It is widely used in data reproduction and image comparison research. This article uses CCC to measure the linear correlation between channels as feature information for posture recognition. Use $x = \{x_1, x_2, \cdots, x_N\}$ and $y = \{y_1, y_2, \cdots, y_N\}$ to represent the time series of two different channels, and calculate the CCC between the two channels according to Eqs. (3)–(5), where $\bar{y}$ is similar to $\bar{x}$ and $s_y^2$ is similar to $s_x^2$:

$$\rho = \frac{2s_{xy}}{s_x^2 + s_y^2 + (\bar{x} - \bar{y})^2} \tag{9}$$

$$\bar{x} = \frac{1}{N} \sum_{n=1}^{N} x_n \tag{10}$$

$$s_x^2 = \frac{1}{N} \sum_{n=1}^{N} (x_n - \bar{x})^2 \tag{11}$$

$$s_{xy} = \frac{1}{N} \sum_{n=1}^{N} (x_n - \bar{x})(y_n - \bar{y}) \tag{12}$$

where $\bar{x}$ and $\bar{y}$ are the average values of x and y. $s_y^2$ and $s_x^2$ are the variances of x and y. $s_{xy}$ is the covariance between x and y.

Then, in this article, the EMG signal collected by the EMG sensor containing M channels is denoted as $X = \{X_1, X_2, \cdots, X_M\}$. Therefore, the CCC of the i-th and j-th channels is represented as

$$P_{X_i X_j} = \rho_{X_i X_j} \tag{13}$$

where $i \neq j$, and $i = 1, 2, \cdots, M - 1$, $j = i + 1$. The linear correlation feature between channels is represented as

$$P = [P_1, ..., P_k, ..., P_{M-1}] \tag{14}$$

where $P_k = \left[\rho_{X_k X_{k+1}}, \rho_{X_k X_{k+2}}, ..., \rho_{X_k X_M}\right]$ and $k = 1, 2, ..., M - 1$.

### Nonlinear correlation characteristics between channels

Mutual information is an essential tool for measuring nonlinear relationships and is widely used in fields such as recommendation systems. This article uses copula mutual information to measure the nonlinear correlation between channels as feature information for pose recognition. The mutual information between sequences X and Y is defined as

$$I(X; Y) = \iint p_{XY}(x, y) \log \frac{p_{XY}(x, y)}{p_X(x) p_Y(y)} \, dx dy \tag{15}$$

where $p_X(x)$ and $p_Y(y)$ are the edge probability density functions of X and Y, respectively, and $p_{XY}(x, y)$ is the joint probability density of X and Y.

On the other hand, the relationship between mutual information and copula entropy is

$$I(X, Y) = -H_c(u, v) \tag{16}$$

where $u = P_X(x)$ and $v = P_Y(y)$. $H_c(u, v)$ is the copula entropy of u and v, $P_X(x)$ and $P_Y(y)$ are the cumulative distribution functions of X and Y, respectively.

For two time series $X = [X_1, X_2]$ with N samples in different channels, perform a probability integral transformation on X based on the empirical distribution function to generate a pseudo observation value $\hat{U}_n = [\hat{U}_{n1}, \hat{U}_{n2}]$, where $n = 1, 2, \cdots, N$. The calculation formula for $\hat{U}_n$ is:

$$\hat{U}_{nl} = \frac{1}{N+1} r_{nl} \tag{17}$$

where $r_{nl}$ is the order statistic of $X_{nl}$ in $\{X_{n1}, \cdots, X_{nl}, \cdots X_{Nl}\}$, $l = 1, 2$.

Finally, the nonlinear correlation features are represented as vectors

$$I = [I_1, ..., I_k, ..., I_{M-1}] \tag{18}$$

where $I_k = [H_{X_k X_{k+1}}, H_{X_k X_{k+2}}, ..., H_{X_k X_M}]$ and $k = 1, 2, \cdots, M - 1$.

### Multi-channel correlation characteristics of multi-channel EMG signals

By using the above method to calculate the linear correlation features between channels and estimate the nonlinear correlation features between channels, a multi-channel EMG

---

**Algorithm 1** MC feature construction method.

Input: Single M-dimensional multi-channel time series sample

Output: MC feature vector $\bar{F}$.

  1: Based on a single M-dimensional data sample $X = [X_1, X_2, \cdots, X_N]$, calculate CCC features between multiple channels:

$$P = \left[ \rho_{X_1 X_2}, \ldots, \rho_{X_1 X_M}, \rho_{X_2 X_3}, \ldots, \rho_{X_2 X_M}, \ldots, \rho_{X_{M-1} X_M} \right]$$

  2: Calculate Copula MI features between multiple channels:

$$I = \left[ H_{X_1 X_2}, \ldots, H_{X_1 X_M}, H_{X_2 X_3}, \ldots, H_{X_2 X_M}, \ldots, H_{X_{M-1} X_M} \right]$$

  3: Construct MC features by combining channel linear and nonlinear correlation features:

$$F = \left[ \rho_{X_1 X_2}, \ldots, \rho_{X_{M-1} X_M}, H_{X_1 X_2}, \ldots, H_{X_{M-1} X_M} \right]$$

  4: Normalize the MC feature vector:

$$\bar{F} \leftarrow norm(F)$$

---

signal multi-channel correlation feature is constructed by combining the two correlation features:

$$F = \left[ P_1, P_2, ..., P_{M-1}, I_1, I_2, ..., I_{M-1} \right] \tag{19}$$

For a dataset containing S samples, S feature vectors with dimension $M(M-1)$ will be obtained, denoted as:

$$F_p = \left[ f_{p,1}, \ldots, f_{p,\frac{M(M-1)}{2}}, f_{p,\frac{M(M-1)}{2}+1}, \ldots, f_{p,M(M-1)} \right] \tag{20}$$

Next, normalize the features of each dimension in the feature vector separately:

$$\bar{f}_{p,q} = \frac{f_{p,q} - f_q}{\sigma_q \sum\limits_{p=1}^{s} f_{p,q}} \tag{21}$$

where $f_q = \frac{\sum\limits_{p=1}^{s} f_{p,q}}{S}$ and $\sigma_q = \sqrt{\frac{\sum\limits_{p=1}^{s} (f_{p,q} - f_q)^2}{S}}$, $p = 1, 2, \cdots, S$, $q = 1, 2, \cdots, M(M-1)$.

Thus, obtaining a normalized feature vector

$$\bar{F}_p = \left[ \bar{f}_{p,1}, \cdots, \bar{f}_{p,\frac{M(M-1)}{2}}, \bar{f}_{p,\frac{M(M-1)}{2}+1}, \cdots, \bar{f}_{p,M(M-1)} \right] \tag{22}$$

The specific method for constructing MC features is shown in Algorithm 1.

Remark 1. There are several reasons for choosing copula MI in our study. Firstly, copula MI has a strong theoretical foundation for measuring nonlinear channel correlations. It is

---

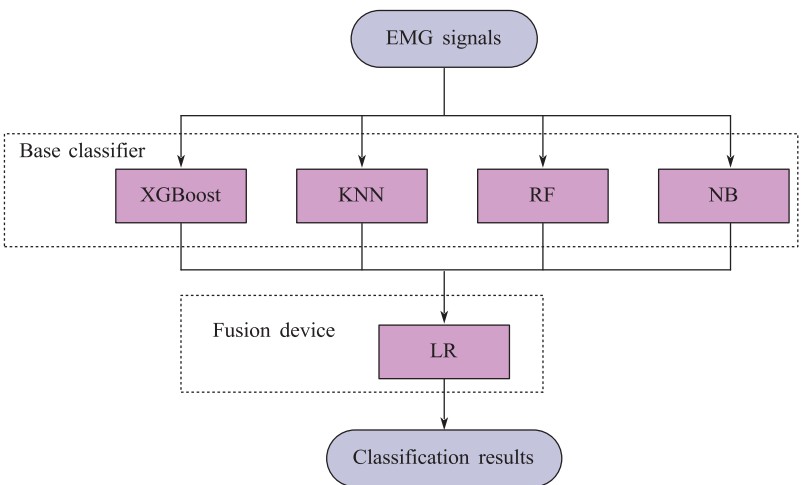

**Figure 4  Block diagram of stacking ensemble learning model.**

directly related to mutual information and can effectively capture complex dependency relationships between variables. Secondly, copula MI demonstrated excellent performance in processing data features in our electromyographic signal analysis. Compared with other methods, it can better adapt to the nonlinearity and non-stationarity of electromyographic signals.

Regarding other nonlinear metrics, such as Pearson correlation or Hilbert Huang transform (HHT) for phase synchronization, although they may have their own advantages in certain situations, they also have limitations. For example, Pearson correlation with phase synchronization may be more sensitive to specific phase relationships and may not fully capture general nonlinear dependencies in our data. HHT may involve more complex calculations, and interpreting the results in the context of our specific problem may not be as simple. Therefore, based on our data's characteristics and our research's requirements, copula MI has been determined as the most suitable nonlinear metric for extracting nonlinear correlated features of electromyographic signals.

### Stacking ensemble learning model

After constructing the MC features of multi-channel EMG signals, the feature vector is input into the classifier for gesture recognition. The traditional classification method identifies the most appropriate feature distribution function within the assumed function space as the basis for classification. However, with the increase in data volume and diversity, especially the influence of individual differences, the distribution of actual data often has uncertainty, which leads to a single classifier's weak generalization ability in multi-classification tasks due to sample uncertainty. In contrast, learning methods integrating multiple classifiers exhibit more significant advantages in high-precision classification tasks.

This article adopts the stacking ensemble learning method. Firstly, the base learner is trained on the initial training set, and then the predicted results of the base learner are used

as a new feature set to train the fusion device. In this stacking framework, extreme gradient boosting (XGBoost), K-nearest neighbors (KNN), Random Forest (RF), and naive Bayes (NB) are selected as base learners, and logistic regression (LR) models are used as fuses. The structural diagram of the model is shown in Fig. 4.

Remark 2. For the KNN algorithm, we conducted a grid search within the range of k-values. We start with a set of [3,5,7,9,11] and evaluate the model's performance using a validation set. The performance indicators are accuracy, precision, recall, and F1 score. Through this process, we found that the k-values of [27,29,31,33,35] produced the best overall performance across multiple datasets.

# EXPERIMENTAL RESULTS

## Experimental preparation

The data used in this article are the leg EMG signals mentioned in *de Freitas & Kohn (2024)*, the biceps EMG signals mentioned in *Khodadadi et al. (2023)*, the hand EMG signals mentioned in *Hao et al. (2023b)*, and the lumbar EMG signals mentioned in *Cheng et al. (2023b)*, *Wang et al. (2024)*, *Hao et al. (2023a)*. The raw data is shown in Fig. 5.

The application was written in Python 3.7, and the experiment was carried out within the PyCharm environment. The machine was set up with an Intel Xeon E5 series processor, 8 GB of RAM, and a 64-bit version of Windows 10. This chapter performs performance evaluation on the test set after the model's training on the training set to assess the recognition performance of the suggested approach. Performance evaluation indicators include precision (P), recall (R), F1, accuracy (Ac), and Matthews correlation coefficient (MCC). The formula for calculating these indicators is as follows:

$$Ac = \frac{TP + TN}{TP + FP + TN + FN} \tag{23}$$

$$P = \frac{TP}{TP + FP} \tag{24}$$

$$R = \frac{TP}{TP + FN} \tag{25}$$

$$Fl = \frac{2PR}{P + R} \tag{26}$$

$$MCC = \frac{TP \times TN - FP \times FN}{\sqrt{(TP + FP)(TP + FN)(TN + FP)(TN + FN)}} \tag{27}$$

where TP is the positive sample that the model anticipates to be positive, The model predicts TN, a negative sample, to be negative, and FP, a negative sample, to be positive. FN is a positive sample that the model anticipated to be negative.

This article superimposes noise with different signal-to-noise ratios (10, 20, 30, 40, and 50 dB) on the original signal to evaluate the signal-to-noise ratio of different methods. The experimental results are shown in Table 1. It can be seen that the improved threshold function proposed in this article has a higher signal-to-noise ratio, indicating that the proposed method can better remove noise.

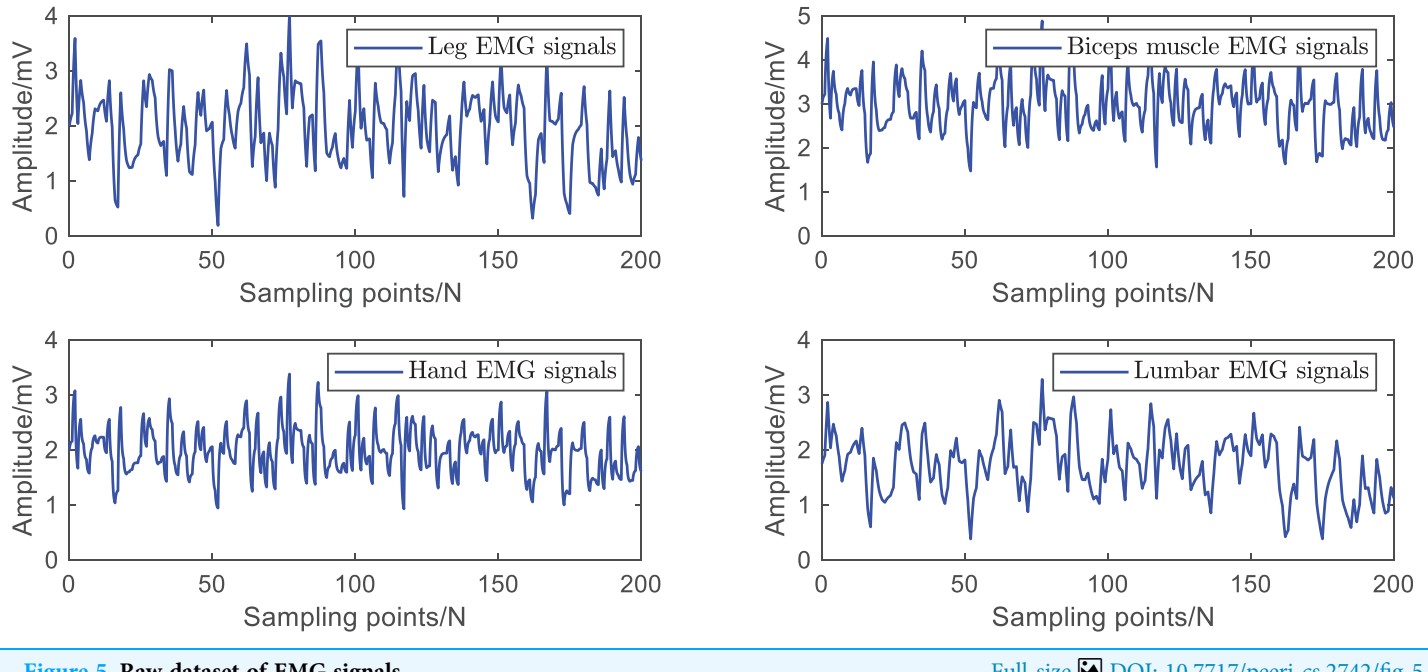

**Figure 5** Raw dataset of EMG signals.

**Table 1 Improved noise performance under different threshold values (dB).**

| Method | 10 dB | 20 dB | 30 dB | 40 dB | 50 dB |
|---|---|---|---|---|---|
| Soft threshold function | 14.8 | 13.5 | 10.9 | 8.7 | 8.2 |
| Hard threshold function | 15.1 | 14.7 | 12.1 | 9.5 | 8.0 |
| Improving threshold function | 20.9 | 19.7 | 17.4 | 16.5 | 16.2 |
| Method | 10 dB | 20 dB | 30 dB | 40 dB | 50 dB |
| Soft threshold function | 14.8 | 13.5 | 10.9 | 8.7 | 8.2 |

## Experimental analysis

Firstly, this article uses a control coefficient of $a = 0.3$ to conduct denoising experiments on the surface EMG signal of the arm. The simulation results are shown in Fig. 6. Using simulation results and signal-to-noise ratio as performance indicators, the higher the signal-to-noise ratio, the better the denoising effect and the better the ability to recover the original signal.

Perform ten-fold cross-validation on each dataset separately, take the average of ten performance evaluation metrics as the result, and calculate the mean of all performance evaluation metrics. The recognition results are shown in Table 2.

Table 2 shows that the recognition accuracy of each dataset has reached over 96%, and the mean values of various evaluation indicators have also exceeded 96%, indicating that the algorithm has achieved good recognition performance. In the final cross-validation, the parameters of the stacking model were set, as shown in Table 3.

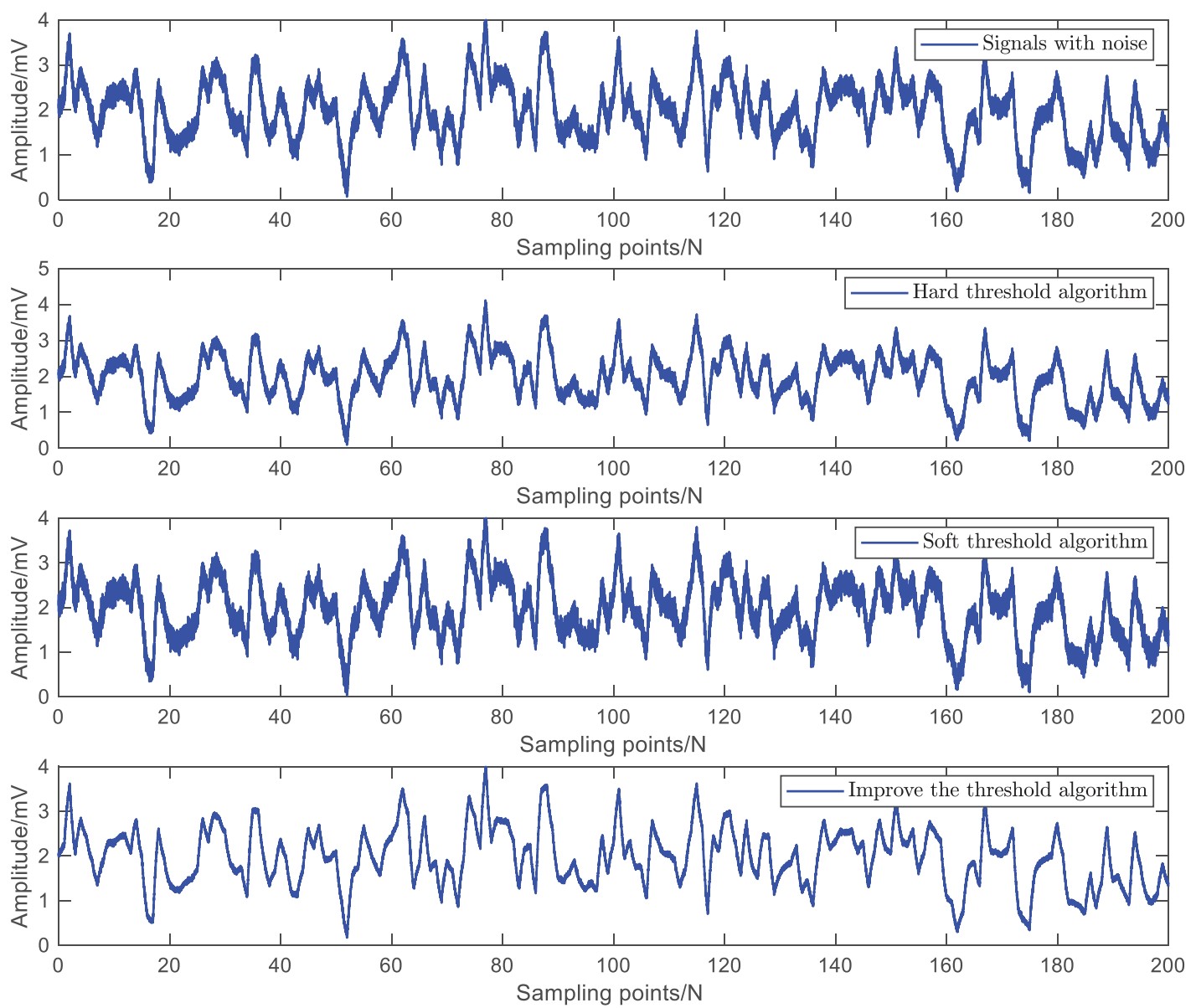

**Figure 6 Simulation results of three threshold denoising methods.**

| Table 2 Recognition results of different datasets. | | | | | |
|---|---|---|---|---|---|
| Data | Ac | P | R | F1 | MCC |
| Leg | 0.9677 | 0.9680 | 0.9633 | 0.9725 | 0.9700 |
| Biceps muscle | 0.9635 | 0.9654 | 0.9681 | 0.9821 | 0.9679 |
| Hand | 0.9738 | 0.9689 | 0.9768 | 0.9689 | 0.9716 |
| Lumbar | 0.9768 | 0.9813 | 0.9719 | 0.9699 | 0.9713 |
| Average value | 0.9706 | 0.9709 | 0.9700 | 0.9734 | 0.9702 |

**Table 3 Stacking model parameters.**

| Parameter name | |
|---|---|
| The number and learning rate of weak learners in XGBoost model | 100, 0.01 |
| The number of nearest neighbors in KNN model | 100 |
| The number and maximum depth of RF model trees | 80, 5 |
| Minimum sample size on leaf nodes | 1 |
| Minimum sample size on branch nodes | 2 |

**Table 4 Recognition results of model parameters tuning.**

| Adjustment parameters | Ac | P | R | F1 | MCC |
|---|---|---|---|---|---|
| Before adjusting parameters | 0.9536 | 0.9369 | 0.9593 | 0.9435 | 0.9410 |
| After adjusting parameters | 0.9768 | 0.9813 | 0.9719 | 0.9699 | 0.9713 |

Parameter settings have a significant impact on model performance. To verify the recognition performance of the stacked model after parameter optimization, this article randomly selected a single dataset for experiments. In the experiment, the parameters of the stacking model were set to default values, and the average of ten performance evaluation metrics was taken as the final evaluation result. The recognition results are shown in Table 4.

Table 4 shows that compared to the model with default parameters, the model with optimized parameters has a higher recognition accuracy, indicating that reasonable parameter settings can effectively improve model performance. Meanwhile, the adjusted parameters have a faster training time.

To verify the effectiveness of the MC feature proposed in this article, it was compared with four commonly used time-domain features: mean absolute value of amplitude (MAV), zero crossing points (ZC), wavelength (WL), and slope change number (SSC). Extract these four time-domain and MC features from all training and testing datasets and complete the classification model training on the training dataset. Then, experiments on the testing dataset will be conducted to obtain the classification results. The average recognition accuracy is shown in Fig. 7.

Overall, the recognition performance of MC features is the best. The recognition accuracy of the four commonly used time-domain features is 78.2% (MAV), 52.8% (ZC), 72.4% (WL), and 58.8% (SSC), respectively. The recognition accuracy of the MC feature is significantly higher than that of these time-domain features, increasing by 41.9 percentage points compared to the ZC feature, reaching 94.7%. Although commonly used time-domain feature extraction is simple and convenient, it is easily affected by changes in force or speed in practical scenarios, resulting in significant fluctuations in feature values and ultimately affecting recognition performance. The MC feature proposed in this study quantifies linear and nonlinear correlations between EMG channels, providing a measure of intermuscular coupling and demonstrating enhanced robustness. Compared to single

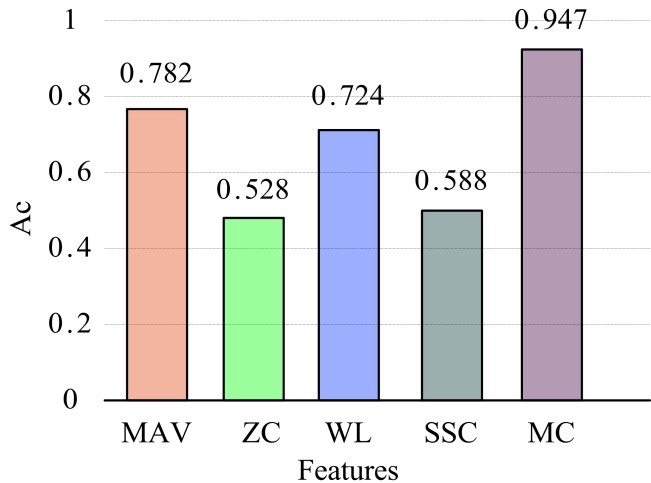

**Figure 7** Average recognition accuracy of different features.

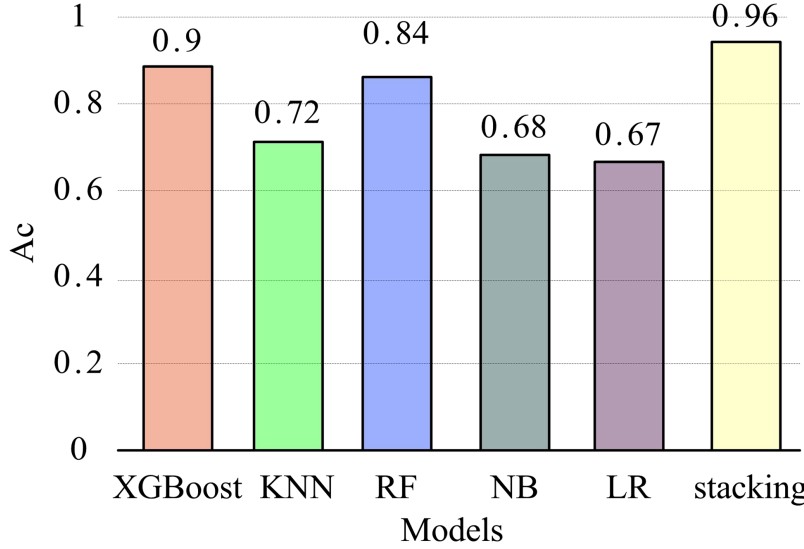

**Figure 8** Recognition accuracy of different models.

time-domain features, MC features effectively leverage inter-channel information, offering a more comprehensive representation of multi-channel EMG characteristics and significantly improving feature recognition accuracy.

To further validate the recognition performance of the stacking model in this article, five commonly used classification models (XGBoost, KNN, RF, NB, LR) were compared with the stacking model to explore their impact on EMG feature recognition. The parameter settings of each model are consistent with those in the stacking model. After extracting MC features from all datasets, the models were trained using the training set and validated on the test set through five-fold cross-validation. The mean and variance of five recognition accuracy measurements were used as evaluation metrics. The experimental results are presented in Fig. 8.

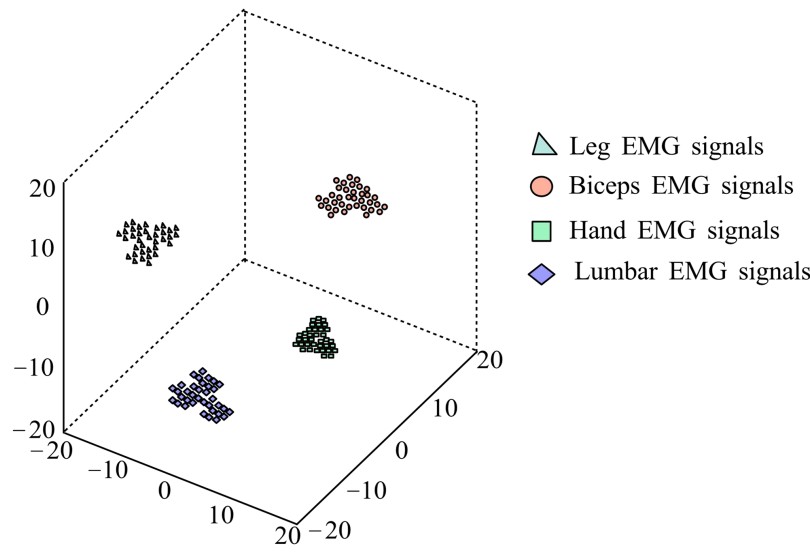

**Figure 9  t-SNE visualization results.**               

Figure 8 shows that XGBoost has the highest accuracy among the four basic models, while the LR model has the lowest recognition accuracy. The recognition accuracy of the stacking model is significantly better than other single classification models, with an average of 96%. This indicates that the stacking model performs better overall and has a higher gesture recognition rate. The reason is that in multi-class classification problems, the representation ability of a single machine learning method is limited, while stacking ensemble learning reduces variance and improves generalization ability by combining the outputs of multiple base classifiers and the learning of fusion devices, thereby improving classification accuracy.

After classification experiments on the data, the MC features extracted from the test set are mapped to three-dimensional space using the t-SNE method for visualization analysis. Figure 9 shows the visualization results of extracting MC features from the test set data. It can be seen that after MC feature extraction, the intra-class distance of various gestures is relatively small, and the inter-class distance is relatively large, showing obvious separability.

To comprehensively analyze the advantages and disadvantages of different feature extraction methods and further demonstrate the superiority of our MC features, the method proposed in this article is compared with power spectral density (PSD) and short-time Fourier transform (STFT) frequency domain features. For PSD, we will calculate the power distribution at different frequencies to capture the frequency characteristics of the signal. For STFT, we will use appropriate window sizes and overlaps to obtain the time-frequency representation of the signal. The experimental results are shown in Table 5.

To verify the algorithm's effectiveness proposed in this article, we removed the CCC and copula MI features separately to evaluate their independent effects. The experimental results are shown in Table 6.

**Table 5 Comparison of experimental results of different different features.**

| Method | Ac | P | R | F1 | MCC |
|---|---|---|---|---|---|
| PSD | 0.8723 | 0.8103 | 0.9078 | 0.7239 | 0.8187 |
| STFT | 0.8472 | 0.8311 | 0.9185 | 0.8285 | 0.8796 |
| MC | 0.9263 | 0.9183 | 0.9413 | 0.9086 | 0.9518 |

**Table 6 Results of ablation experiment.**

| Method | Ac | P | R | F1 | MCC |
|---|---|---|---|---|---|
| Removed the CCC | 0.8523 | 0.7958 | 0.8567 | 0.8423 | 0.8672 |
| Removed the Copula MI | 0.8627 | 0.8078 | 0.9012 | 0.8752 | 0.9011 |
| Proposed method | 0.9263 | 0.9183 | 0.9413 | 0.9086 | 0.9518 |

**Table 7 The performance of the method proposed in this article on various datasets.**

| Datasets | Ac (%) | MCC | AUC-ROC (%) |
|---|---|---|---|
| Arm muscle electrical signal | 96.3 | 0.92 | 98.5 |
| Thigh muscle electrical signal | 95.8 | 0.91 | 97.9 |
| Hand electromyographic signal | 94.7 | 0.89 | 96.8 |
| Lumbar electromyography signal | 96.5 | 0.93 | 98.6 |

The experimental results show that the model using CCC features alone performs better than only copula MI features. Still, the best performance is achieved by combining the MC features, indicating that linear and nonlinear features complement each other in different scenarios.

Finally, this article added the AUC-ROC metric to evaluate the model's performance further, and together with existing metrics (accuracy, MCC), it demonstrated the performance of the stacked model on various datasets. The results are shown in Table 7.

## Limitations of the approach

The proposed method achieves high accuracy in sports behavior prediction, but its generalizability across different sports and movement patterns remains uncertain. The computational complexity of multi-channel feature extraction and ensemble learning may limit real-time applicability, and the reliance on sensor quality introduces variability. Feature selection bias and the lack of interpretability in the stacking model could hinder practical use. Although the approach effectively mitigates noise and crosstalk, its robustness under extreme conditions requires further validation. Additionally, broader comparisons with deep learning models and dynamic adaptation for real-time monitoring could enhance its practical impact.

## CONCLUSIONS

To better meet the needs of daily sports work, this article constructs multi-channel related features of electromyographic signals generated by sports behavior. It uses a stacked ensemble learning model for recognition. Firstly, soft and hard threshold denoising methods and improved threshold denoising methods are used to process noisy signals. Next, the concordance correlation coefficient (CCC) computed the linear correlation features between channels. In contrast, the nonlinear correlation features were estimated using copula-based MI to construct the MC features. Finally, a stacked classification model with XGBoost, KNN, RF, and NB as base classifiers and LR as fusion is used for feature recognition. The experimental results show that this method outperforms 95% in accuracy, precision, recall, F1 value, and Matthews correlation coefficient for four common EMG signals.

In future research, the author will further explore the performance of the MC method in different biological signals and verify its generality and robustness.

### Funding

The authors received no funding for this work.

### Competing Interests

The authors declare that they have no competing interests.

### Author Contributions

- Fengjin Ye conceived and designed the experiments, analyzed the data, prepared figures and/or tables, and approved the final draft.
- Yuchao Zhao performed the experiments, performed the computation work, authored or reviewed drafts of the article, and approved the final draft.
- Zohaib Latif conceived and designed the experiments, performed the experiments, authored or reviewed drafts of the article, and approved the final draft.

### Data Availability

The EEG and EMG data is available at Zenodo: Correia, P., Quintão, C., Quaresma, C., & Vigário, R. (2024). Simulated EEG and EMG data for Reference Phase Analysis evaluation. [Data set]. Zenodo. https://doi.org/10.5281/zenodo.13913999.

The code is available in the Supplemental File.

### Supplemental Information

Supplemental information for this article can be found online at http://dx.doi.org/10.7717/peerj-cs.2742#supplemental-information.

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
