# Peer review of "Research on sports activity behavior prediction based on electromyography signal collection and intelligent sensing channel"

_PeerJ Computer Science, doi:10.7717/peerj-cs.2742_

## Round 0.1 · original submission · Major Revisions

Please carefully revise and resubmit

The paper demonstrates significant advancements over traditional threshold denoising algorithms, including better noise reduction, enhanced signal-to-noise ratio, and greater accuracy in information recognition.

Providing more information about the dataset used for training and testing, such as its size, diversity, and collection protocols, would improve the study's transparency and reproducibility

Including an analysis of the computational efficiency of the proposed algorithm, especially in real-time applications, would be valuable for its practical implementation

Adding a discussion on how the model’s predictions can be interpreted by coaches or athletes would enhance its usability and adoption in practical scenarios.

Reviewer 1 ·

Basic reporting

Overall, this paper provides a valuable contribution to EMG-based sports behavior analysis by integrating signal processing innovations and machine learning techniques. Addressing the following limitations will further enhance the robustness, applicability, and impact of this work.
- In Section 3.2, the improved threshold function adds a control coefficient, but no comparative experiments with modern methods like Non-Local Means (NLM) denoising or adaptive thresholding are shown. Add a comparative analysis with recent denoising algorithms to strengthen your claims.
-
While CCC and Copula MI are useful, other nonlinear metrics like Pearson correlation with phase synchronization or Hilbert-Huang Transform (HHT) could provide additional insights. Justify why Copula MI alone was chosen.
-

-
The manuscript heavily focuses on EMG data. Could the MC feature extraction method be generalized to other bio-signals (e.g., EEG or ECG)? A discussion on generalizability in Section 5 would strengthen the paper's impact.
-
In Section 3.1, the EMG signal frequency range is defined as 5–200 Hz. Verify whether this is consistent across all datasets, as variations in sensors or sampling rates could affect preprocessing results.
-

In the stacking ensemble model, explain how base learners' parameters (e.g., kkk-value in KNN, max depth in RF) were tuned. Including this in Table 2 would clarify the optimization process.

Experimental design

It is unclear how the relative contribution of CCC (linear) and Copula MI (nonlinear) features impacts prediction accuracy. Conduct an ablation study to evaluate these contributions separately in Section 4.2. The manuscript heavily focuses on EMG data. Could the MC feature extraction method be generalized to other bio-signals (e.g., EEG or ECG)? A discussion on generalizability in Section 5 would strengthen the paper's impact.
-
In Section 3.1, the EMG signal frequency range is defined as 5–200 Hz. Verify whether this is consistent across all datasets, as variations in sensors or sampling rates could affect preprocessing results.

Validity of the findings

The comparison with time-domain features (MAV, ZC, etc.) is useful, but frequency-domain features like Power Spectral Density (PSD) or Short-Time Fourier Transform (STFT) could be better baselines. Add these comparisons in Section 4.2.
-

Reviewer 2 ·

Basic reporting

Check the attached review report.

Experimental design

Check the attached review report.

Validity of the findings

Check the attached review report.

Additional comments

Check the attached review report.

Annotated reviews are not available for download in order to protect the identity of reviewers who chose to remain anonymous.

---

## Round 0.2 · Minor Revisions

Dear authors, the reviewers has now commented on your revised article, although many of the comments has been incorporated, but still you need to incorporate few suggestions as mentioned by reviewer2.

Please also improve the language of the paper and resubmit

Reviewer 1 ·

Basic reporting

Now seems fine

Experimental design

Ok

Validity of the findings

Ok

Additional comments

Paper may be accepted for publication

Reviewer 2 ·

Basic reporting

1. Thoroughly improve the grammar and language to make it clear and professional.
2. Add a "Major Contributions" subsection at the end of the introduction section to highlight the key contributions for readers' ease
3. Write a "Limitations of the Approach" subsection at the end of the results section to address the potential shortcomings.

Experimental design

no comments

Validity of the findings

no comments

Additional comments

no comments

---

## Round 0.3 · accepted · Accept

Dear authors
Thank you for your revision and for incorporating the suggestions from the reviewers and editor. The paper has been revised in accordance with reviewers' and mine comments. I am happy to let you know that your manuscript is being recommended for publication. Thank you for your contribution to our journal